# Epidemiologic and Genomic Evidence for Zoonotic Transmission of SARS-CoV-2 among People and Animals on a Michigan Mink Farm, United States, 2020

**DOI:** 10.3390/v15122436

**Published:** 2023-12-15

**Authors:** Ria R. Ghai, Anne Straily, Nora Wineland, Jennifer Calogero, Mary Grace Stobierski, Kimberly Signs, Melissa Blievernicht, Yaritbel Torres-Mendoza, Michelle A. Waltenburg, Jillian A. Condrey, Heather M. Blankenship, Diana Riner, Nancy Barr, Michele Schalow, Jarold Goodrich, Cheryl Collins, Ausaf Ahmad, John Michael Metz, Owen Herzegh, Kelly Straka, Dustin M. Arsnoe, Anthony G. Duffiney, Susan A. Shriner, Markus H. Kainulainen, Ann Carpenter, Florence Whitehill, Natalie M. Wendling, Robyn A. Stoddard, Adam C. Retchless, Anna Uehara, Ying Tao, Yan Li, Jing Zhang, Suxiang Tong, Casey Barton Behravesh

**Affiliations:** 1U.S. Centers for Disease Control and Prevention, Atlanta, GA 30333, USA; ofu9@cdc.gov (R.R.G.);; 2Michigan Department of Agriculture and Rural Development, Lansing, MI 48933, USA; 3Michigan Department of Health and Human Services, Lansing, MI 48909, USA; 4Michigan Department of Natural Resources, Lansing, MI 48909, USA; 5U.S. Department of Agriculture Animal and Plant Health Inspection Service, Washington, DC 20250, USA

**Keywords:** SARS-CoV-2, COVID-19, coronavirus, zoonotic disease, animal, mink, *Neogale vison*, One Health

## Abstract

Farmed mink are one of few animals in which infection with SARS-CoV-2 has resulted in sustained transmission among a population and spillback from mink to people. In September 2020, mink on a Michigan farm exhibited increased morbidity and mortality rates due to confirmed SARS-CoV-2 infection. We conducted an epidemiologic investigation to identify the source of initial mink exposure, assess the degree of spread within the facility’s overall mink population, and evaluate the risk of further viral spread on the farm and in surrounding wildlife habitats. Three farm employees reported symptoms consistent with COVID-19 the same day that increased mortality rates were observed among the mink herd. One of these individuals, and another asymptomatic employee, tested positive for SARS-CoV-2 by real-time reverse transcription PCR (RT-qPCR) 9 days later. All but one mink sampled on the farm were positive for SARS-CoV-2 based on nucleic acid detection from at least one oral, nasal, or rectal swab tested by RT-qPCR (99%). Sequence analysis showed high degrees of similarity between sequences from mink and the two positive farm employees. Epidemiologic and genomic data, including the presence of F486L and N501T mutations believed to arise through mink adaptation, support the hypothesis that the two employees with SARS-CoV-2 nucleic acid detection contracted COVID-19 from mink. However, the specific source of virus introduction onto the farm was not identified. Three companion animals living with mink farm employees and 31 wild animals of six species sampled in the surrounding area were negative for SARS-CoV-2 by RT-qPCR. Results from this investigation support the necessity of a One Health approach to manage the zoonotic spread of SARS-CoV-2 and underscores the critical need for multifaceted public health approaches to prevent the introduction and spread of respiratory viruses on mink farms.

## 1. Introduction

Severe acute respiratory syndrome coronavirus 2 (SARS-CoV-2) is now well understood to be a zoonotic virus, capable of infecting a wide range of mammalian species including companion animals (dogs, cats, ferrets, hamsters), captive exotic animals in zoos, sanctuaries, and aquariums (lions, tigers, snow leopards, gorillas, foxes, otters, and others) and free-ranging wildlife (white-tailed deer, mule deer, and others) [1,2]. The animal species that has perhaps garnered the most global attention, however, are mink (*Neogale vison*) which are farmed for fur or other products. Since the Netherlands reported the first infected mink farm in April 2020 [3], 478 mink farms in 13 countries have reported outbreaks of SARS-CoV-2 to the World Organisation for Animal Health (WOAH), the majority of which occurred in 2020 (422 farms in 11 countries) [2]. These outbreaks have identified that farmed mink are typically exposed to the virus from infected workers, are highly susceptible to SARS-CoV-2 lineages that circulated in early 2020 and 2021, experience morbidity and mortality events resulting from infection, and can transmit the virus among mink, to other animals, and to people [4,5,6,7].

Transmission of SARS-CoV-2 from animals to people has been documented in several species, including hamsters [8], a domestic cat [9], a lion [10], white-tailed deer [11,12], and mink. Mink-to-human transmission has been documented in Denmark [13,14], the Netherlands [4], Poland [15], and Canada [16]. While animal-to-human transmission of SARS-CoV-2 is not unique, the only reported instances of viral adaptation in an animal host, followed by subsequent emergence of viruses with novel mutations in the human population, have occurred in farmed mink [17]. In June 2020 in Denmark, the spike protein change Y453F was initially observed in a cluster of cases in mink and people. Between June and November 2020, Denmark identified an estimated 4000 human sequences with this mutation, believed to have originated through mink-to-mink transmission [14]. In August and September 2020, Denmark discovered a new variant, termed Cluster 5, characterized by 11 amino acid substitutions relative to Wuhan-Hu-1, including three amino acid substitutions in spike (Y453F, I692V, and M1229I), as well as loss of two spike amino acid residues at positions 69 and 70 [18]. Cluster 5 was detected in 12 human cases and on five mink farms in the north Jutland region [14,19]. Following the emergence of novel mutations and variants from farmed mink and corresponding public health concerns, the Danish government ordered the culling of over 17 million mink, unseating the country as a top producer of mink fur globally [14]. Resulting from a number of factors including the emergence of new, more human-adapted variants, Cluster 5 has not been detected since 2020.

The exact number of mink farms in the United States is unknown, although estimates range from 100 to 245 farms (personal communication; [20]). SARS-CoV-2 has been detected in mink on 18 farms; active infection among mink has been detected by real-time reverse transcription PCR (RT-qPCR) on 16 farms in Utah (n = 12), Michigan (n = 1), Oregon (n = 1), and Wisconsin (n = 2), while evidence of previous infection associated with antibodies to SARS-CoV-2 has been detected on one farm in Wisconsin and one farm in an undisclosed state [21].

Here, we describe the only documented outbreak of SARS-CoV-2 on a mink farm in Michigan, which began in September 2020. We report the results of a One Health epidemiologic investigation that had the following objectives: (1) determine the source of initial mink SARS-CoV-2 exposure and the degree of spread within the facility’s mink population; (2) evaluate the risk for further spread of the virus on the farm and surrounding area, including mink-to-mink, mink-to-companion animal, mink-to-wildlife, and mink-to-human transmission; and (3) provide evidence-based recommendations to improve worker safety and limit the spread of SARS-CoV-2 on mink farms.

## 2. Methods

### 2.1. Outbreak Description

On 1 October 2020, the Michigan Department of Agriculture and Rural Development (MDARD) and the Michigan Department of Health and Human Services (MDHHS) received notification of suspected SARS-CoV-2 in mink on a farm. Clinical signs were observed in a number of animals which included nasal discharge, ocular discharge, and epistaxis. Officials in the Michigan Department of Natural Resources (MDNR) were subsequently notified. Above average mortality rates were noted on 27 September among the mink herd, which numbered approximately 17,000 (13 mortalities on 27 September; average 3.75 ± 1.75 SD mortalities 23–26 September; Figure 1). Two deceased mink, sampled 28 September, were sent to Michigan State University for SARS-CoV-2 diagnostic testing. Samples from both mink tested presumptive positive by RT-qPCR on October 2. Samples were forwarded to the US Department of Agriculture National Veterinary Services Laboratories (USDA NVSL) for confirmatory testing, where both were confirmed positive for SARS-CoV-2 by RT-qPCR and sequencing on 8 October. On 9 October, officials from MDARD visited the affected mink farm and instituted a quarantine on movement of mink and their products off the farm to prevent the spread of SARS-CoV-2 from the farm to surrounding communities or wildlife.

### 2.2. One Health Field Investigation

On 6 October, officials from the local public health department visited the mink farm to collect samples from farm staff for SARS-CoV-2 testing. At the request of Michigan officials, a field team consisting of three veterinary epidemiologists and one laboratory animal resident from the Centers for Disease Control and Prevention (CDC) deployed and were joined by a field veterinarian from MDARD on the mink farm from 13 to 16 October. This field investigation used a One Health approach that consisted of a facility assessment, animal health assessment, mink sample collection, companion animal sample collection, farm staff sample collection, and farm staff interviews. Wildlife sample collection was also conducted by USDA and MDNR officials in areas adjacent to the mink barns.

All on-farm animal sampling was conducted under the CDC Institutional Animal Care and Use Committee’s approval (protocol 3104BARMULX). Wildlife sampling was conducted as part of operational damage management. All animal work was conducted in collaboration with, and under the approval of, MDHHS, MDARD, and MDNR.

#### 2.2.1. Facility and Animal Health Assessment

An informal facility assessment was conducted to establish barn layouts for sampling, assess facility ventilation, identify the presence and distribution of biosecurity signage, and observe biosecurity practices, including the use of personal protective equipment (PPE).

An informal animal health assessment was conducted to determine the current health of the mink herd, history of disease, and frequency of new mink introductions.

#### 2.2.2. Mink Sample Collection

A systematic random sampling approach was used to sample 1% of the living mink herd distributed throughout the facility, with a target sample size of 150 mink. Half of all sampled mink were sedated with 0.03–0.04 mg/kg dexmedetomidine and 5 mg/kg ketamine injected intramuscularly to permit blood sample collection. After reaching an acceptable level of sedation, mink were taken to a clean processing area for blood sample collection. Oral and rectal samples were collected using plastic-handled polyester-tipped swabs; nasal samples were collected with metal-handled mini polyester-tipped swabs. All swabs were cut with clean scissors prior to placing into a sterile 2 mL tube containing 0.5 mL viral transport medium (VTM). Up to 3 mL of blood was collected from the jugular vein into a sterile vacutainer serum separator tube and left for <1 h until serum separation. Sera were then removed and aliquoted into sterile 2 mL tubes.

The other half of all sampled live mink were collected unsedated to reduce sampling times by restraining the animals behind the shoulder and at the base of the tail. Nasal, oral, and rectal swabs were collected in the same manner as for sedated mink.

Up to five recently deceased mink per day were sampled opportunistically by collecting nasal, oral, and rectal swabs in the manner described above. Up to 3 mL of intracardiac blood was also collected for serum, which was processed in the manner described above.

#### 2.2.3. Companion Animal Sample Collection

There were no companion animals that lived on the farm. All companion animals (dogs and cats) owned by farm employees were sampled unsedated, using physical restraint when necessary. Nasal, oral, and rectal swabs were collected in the same manner as described above for mink. Up to 3 mL of blood was collected via the cephalic or lateral saphenous veins in dogs and from the jugular or medial saphenous veins in cats and processed in the same manner as described above.

All samples were kept in a cooler with ice packs until they could be placed in liquid nitrogen at the end of each day. Frozen samples were shipped on dry ice and then stored at −80 °C until testing.

#### 2.2.4. Wildlife Sample Collection

Free-roaming wild mammals were trapped within a 1 km radius of the farm over a five-day period from 16 to 22 October 2020. Mesocarnivores were targeted using Tomahawk live animal traps baited with sardines and/or cat food. Most traps were placed along forest edges or windbreaks. For each animal, a serum sample and oral, nasal, and rectal swabs were collected.

Swabs were stored in cryovials filled with 0.5 mL brain heart infusion (BHI) growth medium. All samples were kept on ice and shipped to USDA NVSL within 24 h for laboratory testing.

#### 2.2.5. Staff Sample Collection

Nasopharyngeal swabs were collected from consenting farm staff on October 6 using plastic-handled, polyester-tipped swabs that were cut with clean scissors and placed into a sterile 2 mL tube containing 0.5 mL VTM.

On 19 October, health officials from the local public health department visited the farm to collect up to 3 mL of blood from the median cephalic, cubital, or basilic veins of consenting farm staff into a sterile vacutainer serum separator tube. The blood was left for <1 h until serum separation. Sera were then removed and aliquoted into sterile 2 mL tubes.

All samples were kept in a cooler with ice packs until they could be placed in liquid nitrogen. Frozen samples were shipped on dry ice and then stored at −80 °C until testing.

#### 2.2.6. Staff Interviews

Interviews were conducted in English or Spanish with mink farm staff, defined as individuals working (paid or unpaid) or living on-farm. Interviews collected information on previous or current illnesses, potential exposures to SARS-CoV-2 both on the farm and outside of farm duties, activities with mink that may have facilitated work-related exposure (e.g., handling, feeding, cleaning), handwashing practices and use of PPE, and other occupational data such as the method of commuting to work or additional work arrangements [6].

#### 2.2.7. Training and Outreach

Following facility assessments, animal health assessments and interviews, basic training specific to the scenario was provided in English and Spanish. Training included cleaning and disinfection best practices, had hygiene, and how to don and doff PPE.

### 2.3. Diagnostic Testing

#### 2.3.1. On-Farm Animal Sample Testing—RT-qPCR, Serology, Sequencing

All farmed mink and companion animal samples collected during the One Health field investigation were sent to CDC for RT-qPCR, sequencing, and serologic testing. RNA extraction and RT-qPCR followed previously published methods [22,23].

Serological testing used a species-independent, chemiluminescence-based protein complementation assay (‘mix-and-read’) to detect antibodies against the receptor-binding domain of the viral spike protein [24]. Specificity of the assay and signal cut-offs have been previously established for mink, dogs, and cats [6,7,25].

Data on mink age (juvenile or adult), fur color (black or white), housing conditions (paired or single), geographic location within the facility, and sample collection with or without sedation were used to identify a representative sample of RT-qPCR positive samples to be selected for sequencing. Among these, samples with the lowest cycle threshold values, making them commensurate to sequencing, were sent for sequencing. Samples were subjected to a combination of Illumina sequencing and Sanger sequencing following previously published protocols [26]. Consensus sequences were generated with IRMA [27].

#### 2.3.2. Wildlife Sample Testing

All wildlife samples were sent to USDA NVSL for testing by RT-qPCR using previously published methods [22]. Serum samples were tested by virus neutralization assay using previously published methods [28].

#### 2.3.3. Human Sample Testing—RT-qPCR, Serology, Sequencing

Clinical samples were tested by RT-qPCR following previously published methods after extracting RNA with the Kingfisher Flex platform [22,23].

Positive samples were reflexed for whole genome sequencing and prepared using QIAGEN QIAseq SARS-CoV-2 Primer Panel assay and Illumina DNA Prep prior to sequencing on Illumina MiSeq platform. Consensus sequences and quality checks were performed with nCovNGS pipeline developed at MDHHS. The pipeline utilizes bwa, samtools, and ivar to complete mapping of reads and consensus generation. Sequences with >90% genome coverage are uploaded to the Global Initiative on Sharing All Influenza Data (GISAID) with a de-identified sequencing identifier.

### 2.4. Data Analyses

#### 2.4.1. Statistical Analyses

Data were compiled in Microsoft Excel (Microsoft Corporation, version 2303, 2018). Pairwise comparisons using Fisher’s exact test were conducted to determine if test positivity by oral, nasal, and rectal swabs was associated with mink fur color (black or white), age (juvenile or adult), housing status (paired or individual), location (barn 1 or 2), or sample collection method (sedated or unsedated) in OpenEpi (version 3.01) [29].

Linear regression models were developed in SAS to determine whether a linear relationship existed between the Ct values of oral and nasal samples among the following groups: total population, sedated mink, unsedated mink, alive mink, and dead (deceased or euthanized) mink.

#### 2.4.2. Genomic Analyses

Study sequences were included in a phylogenetic analysis alongside SARS-CoV-2 sequences obtained from human specimens collected in Michigan and released on GISAID by 25 January 2021, where they had collection dates and were marked as high-quality sequences. Phylogeny was inferred using maximum likelihood analyses implemented in Treetime using the Nextstrain pipeline [30].

### 2.5. Release from Quarantine

To release live mink from quarantine, SARS-CoV-2 infection status among the mink herd was evaluated on 4 and 25 January 2021 by collecting oral swabs from live, unsedated mink randomly selected based on distribution within the facility. Samples were collected using plastic-handled polyester-tipped swabs placed in 3 mL tubes of BHI. Samples were sent to USDA NVSL for testing by RT-qPCR using previously published methods [22].

## 3. Results

### 3.1. Facility and Animal Health Assessment

The mink farm housed the entire mink herd in two barns, where mink were either paired together in wire cages or housed individually. Prior to the SARS-CoV-2 outbreak, the mink herd on the farm consisted of approximately 17,000 animals, of which 4500 were adult females, and 12,500 were juveniles. The farm only kept mink that were white or black in fur color. This herd was established in 2018 with the introduction of 4500 pregnant females; a previous herd was positive for Aleutian disease and was therefore culled and pelts were harvested. The last introduction of live mink onto the farm prior to the SARS-CoV-2 outbreak was in April 2020.

An estimated 21.3% (n = 3621) of the herd was lost due to the SARS-CoV-2 outbreak, with an estimated 20% population loss among juveniles and 25% loss among adult females. Heightened mortality in the mink herd was observed between 27 September and 10 October; average mortality was 201 (±210.5 SD) animals per day during this time, with a peak mortality of 633 animals lost in a single day (Figure 1). Most mink displayed clinical signs including coughing and ocular discharge during the period of heightened mortality. By October 11, mortality rates had fallen below 10 animals per day. During the One Health field investigation conducted October 13–16, most mink had recovered from illness, although ocular discharge, nasal discharge, and diarrhea were observed in some individuals.

### 3.2. Animal Sample Diagnostic Testing

A total of 165 mink were sampled (Table 1). Nasal, oral, and rectal samples were successfully collected and tested from 159 mink (Table 2). All but one mink had one or more samples that were RT-qPCR positive for SARS-CoV-2 (99%; Table 2). Nasal sample Ct values ranged from 19.3–39.4, oral from 17.1 to 38.5, and rectal from 24.1 to 38.5. Serum was successfully collected and tested from 82 sedated mink; 100% were antibody-positive for SARS-CoV-2 (Table 2).

Three companion animals were owned by farm employees and sampled—two dogs (a 3-year-old female German Shepherd, and a 4-month-old male Cocker Spaniel mix), and one cat (a young adult female domestic short hair). The dogs primarily remained indoors, while the cat lived outdoors. All animals were reportedly healthy in the 30 days prior to sampling. None of these animals were RT-qPCR positive for SARS-CoV-2, although one dog, the German Shepherd, was antibody-positive (Table 2). This dog was owned by Staff 1 (Figure 1), who tested positive for SARS-CoV-2 on 6 October.

Thirty-one wild mammals were captured or collected across the five-day trapping period. These included fourteen Virginia opossums (*Didelphis virginiana*), thirteen raccoons (*Procyon lotor*), one Eastern cottontail rabbit (*Sylvilagus floridanus*), one groundhog (*Marmota monax*), one wild mink, and one Norway rat (*Rattus norvegicus*). The mink was a wild, road-killed individual collected approximately 1 km north of the farm. None of the samples collected from wildlife were positive for viral RNA or antibodies to SARS-CoV-2.

### 3.3. Statistical Analyses

There were no statistical differences in test positivity of mink based on fur color (black or white), age (juvenile or adult), housing status (paired or individual), location (barn 1 or 2), or sample collection method (sedated or unsedated). In linear regression models, oral and nasal Ct values had a statistically significant positive relationship in all groups (total population: *p* < 0.001, slope 0.38, CI −0.25–0.5, R^2^ 21%), suggesting that on average, oral Ct values can predict nasal Ct values in a population, but are imprecise at the individual level.

### 3.4. Staff Interview and Diagnostic Testing

Seven staff worked on the mink farm in the month preceding and during the outbreak, including the farm owner/operator. On 27 September, the first day of heightened mink mortalities, three staff (Staff 1, 4, and 5; Figure 1) reported symptoms consistent with COVID-19, including cough, runny nose, loss of taste/smell, myalgia, chills, and subjective fever.

Two out of seven (29%) staff were positive for SARS-CoV-2 on nasopharyngeal samples collected on October 6, one of whom was symptomatic and had reported symptoms consistent with COVID-19 on 27 September, and another that never reported experiencing symptoms (Staff 1 and 2, respectively; Figure 1). Ct values for these individuals were 15 and 18. During interviews, both individuals reported working in the mink barns, with duties that included feeding mink, removing dead animals, and cleaning cages. Neither staff member reported wearing any PPE prior to the outbreak. Staff were not observed wearing PPE during the field investigation.

On 19 October, blood was collected from six of seven staff; Staff 2 declined further testing. Four of six staff were antibody-positive for SARS-CoV-2 (Staff 1, 4, 5, and 6). Staff 6, on the other hand, did not report previous COVID-19 or a history of symptoms consistent with COVID-19 (Figure 1). Of note, all diagnostic results predated the availability of human vaccines for COVID-19, and therefore are considered indicative of true infection.

### 3.5. Recommendations to Improve Worker Safety and Limit the Spread of SARS-CoV-2 on the Mink Farm

Recommendations were categorized into those intended to control further spread of SARS-CoV-2 on the mink farm and to prevent virus reintroduction.

Recommendations to control the current outbreak included increasing facility ventilation, restricting access to the farm, designating people to work in specific areas, minimizing resource and equipment sharing between barns, disposing of leftover feed, and taking care of healthy animals before sick animals. Staff were trained on cleaning and disinfection methods to limit viral spread through contaminated objects, and PPE donning and doffing. PPE, including a respirator (N95), dedicated outerwear, eye protection (face shield or goggles), and gloves were recommended for any staff working with or near animals on the farm.

Recommendations to prevent future SARS-CoV-2 outbreaks on the farm included providing leave to sick staff or, if no one else could perform their duties, requiring sick staff to always wear an unvented respirator (N95), avoid unnecessary interactions, and limit time in the barn or around animals. Facemasks, in addition to protective outerwear and gloves, were recommended during pelting and processing. Continued use of footbaths and the adoption of EPA-approved disinfectants was encouraged.

### 3.6. Sequencing and Genomic Analyses

Genomic analyses included a total of 71 whole-genome sequences from this study: two mink samples collected on 28 September (used to confirm the mink farm outbreak), sixty-five mink samples collected during the field investigation 13–16 October, two mink samples collected November 30 to assess ongoing presence of SARS-CoV-2 on the farm, and two human samples from farm staff sampled on October 6.

All mink and human sequences associated with the farm outbreak clustered together in a monophyletic clade identified as PANGO lineage B.1. This cluster was 30–35 SNPs different from the reference sequence Wuhan-Hu-1 and was divergent from the 2383 human community sequences selected from Michigan (Figure 2), with at least 40 SNP differences between the clade and the most closely related Michigan community sequence (Figure 3). All sequences in the cluster contained a conserved set of 16 amino acid mutations spread across the genome and one 3 base pair deletion in spike (Table 3). Notably, two spike mutations, F486L and N501T, detected in other mink farm outbreaks in the Netherlands [31], were present in all mink sequences (Table 3). Thirty-seven mink sequences, including those from the September and November collection dates, also contained a 12 base pair insertion in ORF1a that had not been detected previously in any available sequences (Table 3). This insertion was not present in the two sequences from farm staff nor the remaining thirty-one mink sequences from this study. One staff sequence (EPI_ISL_614191) was identical (aside from the 12 base pair insertion in ORF1a) to one of the September mink sequences (USA/MI-028629-001/2020); the other (EPI_ISL_614176) differed by two SNPs.

Genomic analysis also identified one individual from the Michigan background community dataset that clustered closely with the mink outbreak clade. This individual’s sequence, collected on 29 November 2020 (over two months after the start of the mink outbreak), had over 30 SNP differences compared to mink outbreak sequences but possessed F486L and N501T mutations (Figure 4). Following this observation, a retrospective epidemiologic investigation was launched by MDHHS on CDC’s request, which determined this individual had no known direct or indirect association with the affected mink farm.

Phylogenetic analysis indicated viral evolution within the mink outbreak clade through time. Samples collected in September and October differed by 0–3 SNPs, while the two mink sequences collected in November differed by 2–8 SNPs relative to September. Furthermore, Nextstrain molecular clock estimation indicated a faster pace of evolution within the clade than in the broader Michigan community, with the clade rate estimate being 35.9 and the Michigan community estimate being 23.0 substitutions per year (Figure 4).

### 3.7. Release from Quarantine

A total of 63 samples were collected from live mink on 4 January 2021. One sample (1.5%) tested positive by RT-qPCR. Resultingly, the herd was sampled again on 25 January 2021, where another 63 samples were collected. No samples were positive for SARS-CoV-2 at this time point. On 5 February 2021, the quarantine was released for live mink on the facility based on no RT-qPCR positive mink detections, no evidence of clinical signs consistent with SARS-CoV-2, and no increases in mink mortality above baseline levels reported in the subsequent 21 days.

## 4. Discussion

The results of this One Health investigation provide evidence of human to mink, mink to mink, and potentially mink to human zoonotic transmission of SARS-CoV-2. None of the mink farm staff disclosed suspected or known COVID-19 prior to the start of the outbreak; therefore, the specific source of mink herd exposure is unknown. However, the most likely hypothesis is that mink were exposed to SARS-CoV-2 from an asymptomatic person, likely a staff member, on the farm. Since there was an antibody-positive staff member who did not report symptoms and did not test positive for SARS-CoV-2, this individual may have served as an asymptomatic source of infection (Staff 6; Figure 1). This hypothesis is also supported by previous epidemiologic studies, which have identified SARS-CoV-2 infection in animals following close contact with a person with COVID-19 [32,33].

As with other mink farm outbreaks, this investigation also suggests that mink can efficiently spread the virus among their population [6]. Epidemiologically, the outbreak was explosive within the mink herd, resulting in widespread clinical signs and mass mortalities within the same week of initial detection. Exponential increases in infection suggests multiple infected mink were transmitting the virus among the herd. Genomic data also support mink-to-mink transmission; sequences from mink samples collected in November possessed more mutations than those collected in September, and the rate of mutation in the mink outbreak clade was higher than in the broader Michigan community. Both indicate rapid viral evolution resulting from a host shift event to mink.

Some studies have found evidence of infection in other animal species associated with a mink farm outbreak, including cats and dogs living or roaming on the farm, and escaped or wild mink living adjacent [7,34,35]. Here, we discovered that one dog, owned by a mink farm employee with COVID-19 during the investigation, was antibody-positive for SARS-CoV-2. However, we were unable to determine whether this exposure was linked to mink. Similarly, we found no evidence of active infection or previous exposure to SARS-CoV-2 in wildlife trapped nearby.

Finally, both epidemiological and genomic evidence suggests that mink-to-human transmission may have occurred during this outbreak. Our timeline indicates that three staff members (Staff 1, 4 and 5; Figure 1) developed symptoms consistent with COVID-19 on 27 September, the same day that respiratory clinical signs and heightened mortalities were reported in the mink herd. Staff 1, who reported symptoms consistent with COVID-19, and Staff 2, who did not report symptoms, both tested positive for SARS-CoV-2 9 days later. The presence of widespread respiratory signs and heightened mortalities in the mink herd on 27 September indicates likely viral circulation for several days prior, which is supported by the dissimilarity between the earliest mink sequences and those from the broader Michigan community. Both positive staff members were likely exposed to infected mink during this time, having reported working in mink barns on tasks that included feeding mink and cleaning enclosures without the use of PPE. Further, since sequences from SARS-CoV-2 positive staff members differed from mink sequences collected in September by 0–2 SNPs, infection of these staff through unrelated community exposure is less likely. However, our epidemiologic evidence cannot rule out the possibility that these COVID-19 positive staff were infected by the same asymptomatic individual that exposed the mink.

Genomic evidence provides further support for the hypothesis that COVID-19 positive staff were infected by mink. Sequences from both staff included the F486L and N501T mutations, which are believed to be associated with viral adaptation to mink and have been detected in other mink farm outbreaks in the Netherlands [31,36,37]. These mutations are relatively uncommon in human SARS-CoV-2 sequences; between January 2020 and the start of the mink farm outbreak on 27 September 2020, 49 out of 114,598 (0.04%) US human sequences available on GISAID possessed the N501T mutation, and only 2 (0.002%) possessed the F486L mutation. There were no human sequences reported in the United States with both mutations in this timeframe. In the Netherlands, the emergence of the F486L mutation was coincident with an increased incidence of SARS-CoV-2-positive mink farms, and associated with more rapid evolution and greater viral spread in mink [31]. If the F486L amino acid mutation is advantageous to spread in mink populations, it is notable that it arose via different nucleic acid substitutions in the Netherlands mink farm outbreaks (T23045C) than the outbreak described here (T23047G), providing further support that this mutation is positively selected for in mink. Regardless of potential fitness advantages associated with the F486L and N501T mutations, their detection in staff sequences suggests these individuals were infected by a virus that had been passaged through mink. This hypothesis is limited, however, by the unavailability of sequences from the individual that initially exposed the mink, and by the lack of robustness of community sequences from routine strain surveillance in mid-to-late 2020. Without comprehensive background data, it is difficult to determine whether community transmission may have occurred, resulting in the detection of the Michigan individual whose sequence clustered with but was not epidemiologically linked to, the mink farm outbreak.

Evidence of mink-to-human transmission is not surprising given previous detections on mink farms in Denmark [13,14], the Netherlands [4], and Poland [15]. Mink farms are also known to be highly contaminated with SARS-CoV-2 RNA; a study of infected Dutch mink farms identified high levels (1000–10,000 copies/m^3^) in airborne personal inhalable dust, signaling the considerable occupational hazard associated with mink farm work during SARS-CoV-2 outbreaks [38]. More recently, while no human infections were reported, the circulation of highly pathogenic avian influenza A(H5N1) on a Spanish mink farm has illuminated that the occupational risk of contracting respiratory disease on mink farms is not limited to SARS-CoV-2 [39]. Taken together, mink appear to be uniquely susceptible to respiratory zoonotic diseases, and intensive farming creates scenarios that are favorable for bidirectional zoonotic transmission. Taking a One Health approach is therefore important to prevent, detect, and respond to these events, while fostering stronger biosecurity, worker safety, and ongoing surveillance are necessary interventions to protect the health of people and animals in this farming system from SARS-CoV-2 and other emerging and zoonotic disease threats.

## 5. Conclusions

The results of this investigation support that SARS-CoV-2, perhaps especially the lineages circulating in 2020, are highly contagious to mink. Intensive farming, where high densities of animals exist in perpetual close proximity to one another, exacerbates natural species susceptibility, creating highly contaminated environments that result in widespread transmission among mink herds and to other susceptible species, including people. We also discovered that irrespective of clinical signs or animal signalment, all but one mink sampled during field investigations, which occurred after the height of peak mortalities, tested positive for SARS-CoV-2, indicating that practices to separate sick animals from apparently health animals during a SARS-CoV-2 outbreak may be ineffective at mitigating viral spread. These results support current US government messaging, which states that the risk of SARS-CoV-2 infection from animals is considered low for most people in the United States, but that individuals working on mink farms may be at higher risk.

Mink farming in the United States is not regulated by the USDA nor most state departments of agriculture. As a result, there is no mechanism for farmers to collect indemnities for losses associated with disease outbreaks. Despite these challenges, our results and those of other mink farm investigations show that strong collaborative relationships are necessary to mitigate both public health and animal health risks associated with SARS-CoV-2 and other zoonotic respiratory virus infections among farmed mink. On a positive note, fewer SARS-CoV-2-affected mink farms have been reported each year since 2020. Declines in mink susceptibility to variants more adapted to human–human transmission may partly explain this trend. However, One Health actions, including engaging industry partners at all levels, promoting human vaccination, mink vaccination, heightened biosecurity, and use of PPE may also have contributed to case declines on mink farms. Successes associated with preventing SARS-CoV-2 introductions on mink farms should not, however, result in complacency. In late 2022 and early 2023, a new a SARS-CoV-2 lineage was discovered circulating cryptically on Polish mink farms. The identified lineage possessed over 40 SNP differences from the most closely related human sequences, as well as the F486L and N501T mutations, indicating that despite progress, the threat of SARS-CoV-2 on mink farms has not abated [40]. Continued heightened vigilance and plans for swift action remain critical to ensure this human–animal interface remains secure from SARS-CoV-2 and other emerging and zoonotic disease threats.

## Figures and Tables

**Figure 1 viruses-15-02436-f001:**
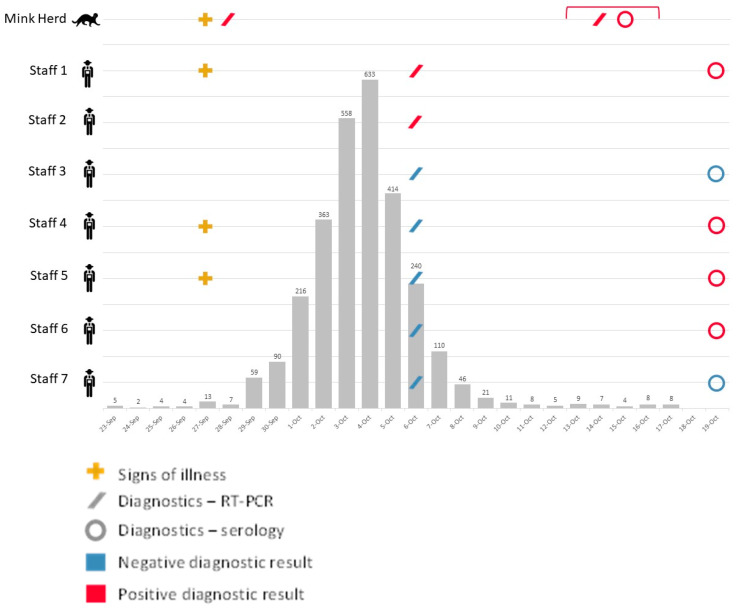
Timeline of events. Epidemic curve of mink mortalities * in grey, overlaid by onset of illness (crosses), sample collection, and diagnostic results (slashes and open circles) of the mink herd and seven staff present on a Michigan mink farm confirmed positive for SARS-CoV-2, 23 September–19 October 2020. Red bracket indicates dates of One Health field investigation, 13–16 October. * Mink mortality data were only collected until 17 October 2020.

**Figure 2 viruses-15-02436-f002:**
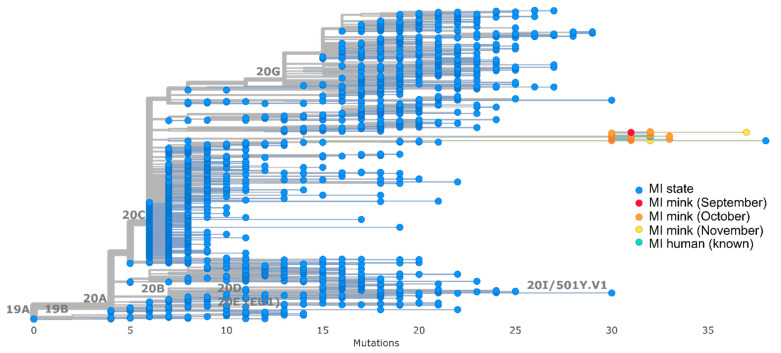
Nextstrain divergence tree where blue tree tips indicate background Michigan human community sequences selected March 2020 to January 2021, red, orange, and yellow are mink sequences from September, October, and November, respectively, and teal are outbreak-associated human sequences.

**Figure 3 viruses-15-02436-f003:**
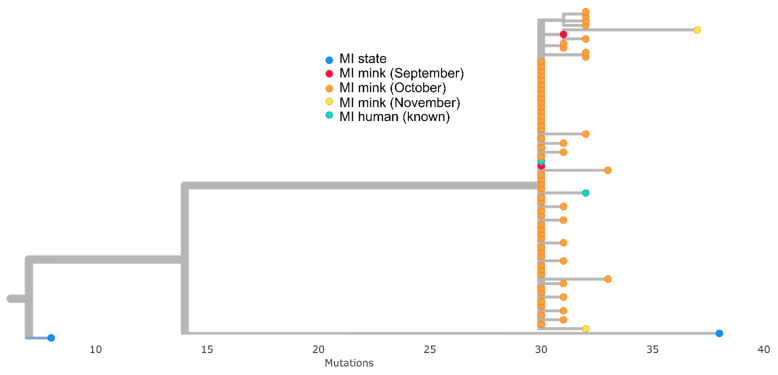
Closeup view of mink clade and associated sequences blue tree tips indicate Michigan human community samples taken March 2020 to January 2021, red, orange, and yellow are mink sequences from September, October, and November, respectively, and teal are outbreak-associated human sequences.

**Figure 4 viruses-15-02436-f004:**
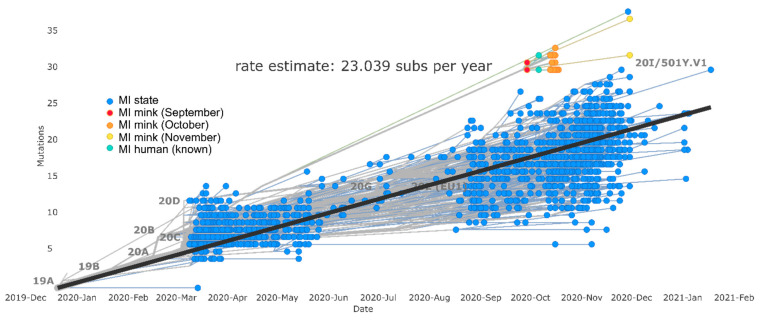
Nextstrain molecular clock tree where blue tree tips indicate Michigan human community samples taken March 2020 to January 2021, red, orange, and yellow are mink sequences from September, October, and November, respectively, and teal are outbreak-associated human sequences.

**Table 1 viruses-15-02436-t001:** Demographics of farmed mink that were sampled 13–16 October. Sedation numbers are for alive animals only.

	Barn		Age, Sex			Color Morph		Housing		Status		Sedation	
	1	2	Adult Female	Juv. Female	Juv. Male	Black	White	Paired	Single	Alive	Dead	Unsedated	Sedated
No. mink (N = 165)	79	86	53	59	52	85	80	108	57	152	13	70	82

**Table 2 viruses-15-02436-t002:** Diagnostic testing results by sample type for all animals tested on mink farm, including total numbers tested and the number and percentage positive for SARS-CoV-2. Nasal, oral, and rectal samples were tested by RT-qPCR, while serum samples were tested by a mix-and-read assay.

Animal (N)	Nasal		Oral		Rectal		Serum		Total	
	Total	Pos. (%)	Total	Pos. (%)	Total	Pos. (%)	Total	Pos. (%)	Total	Pos. (%)
Mink (165)	158	145 (92%)	159	148 (93%)	158	134 (85%)	82	82 (100%)	557	509 (91%)
Dog (2)	2	0 (0%)	1	0 (0%)	2	0 (0%)	2	1(50%)	7	1 (14%)
Cat (1)	1	0 (0%)	1	0 (0%)	1	0 (0%)	1	0 (0%)	4	0 (0%)

**Table 3 viruses-15-02436-t003:** Mutations present in the 71 mink (n = 69) and human (n = 2) sequences associated with the Michigan mink farm outbreak.

Gene	Presence in Sequenced Samples (N = 71)	Mutation, Insertion, or Deletion
ORF1a	All (71)	T265I
	All (69) *	V561A
	37 mink	4 aa insertion at 2037
	All (71)	Q3777R
	All (71)	T4164I
	All (71)	T4174I
ORF1b	All (71)	P314L
	All (71)	T2537I
Spike	All (70) *	G142D
	All (69) *	Y144del
	All (71)	F486L
	All (71)	N501T
	All (71)	D614G
ORF3a	All (71)	Q57H
	All (71)	L219V
Nucleocapsid	All (71)	D103Y
	All (71)	R191L
Envelope	All (71)	C44Y

* Sequence data for one or both human sequence(s) are unavailable at indicated position.

## Data Availability

Complete or near-complete genome sequences of SARS-CoV-2 obtained in this investigation are available at Global Initiative on Sharing All Influenza Data (GISAID) and GenBank. Accession numbers and metadata for these sequences are available in Appendix A. Additional information or de-identified data may be made available to researchers who submit a methodologically sound proposal to the corresponding author.

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
