# Peer review of "Epidemiologic and Genomic Evidence for Zoonotic Transmission of SARS-CoV-2 among People and Animals on a Michigan Mink Farm, United States, 2020"

_viruses, 2023, doi:10.3390/v15122436_

Round 1

Reviewer 1 Report

Comments and Suggestions for Authors

This paper is a multiple author and multiple agency study of the virology and epidemiology of a SARS CoV2 outbreak on a mink farm in Michigan.  The study investigates the only documented outbreak of SARS CoV2 on a mink farm in Michigan.  The study examined the outbreak in mink and the potential spread of the mink virus to farm workers and possibly vice versa.

The study is well designed and appropriate virology and serological samples were obtained from both minks and the farm workers.  Essentially, it was determined that the SARS CoV2 strain in the Michigan has similar mutations detected on similar viral strains isolated in Denmark and the Netherland with some sequence differences in the Michigan strains.  The study provides good epidemiological, virological and sequence data that the farm workers were likely infected with the same SARS CoV2 virus in the mink.

The authors also did a very interesting comparison of the sequences from the mink and persons on the mink farm to the community of sequences from the population in Michigan.  Not surprisingly, the sequence of the viruses from the mink farm and the associated farm workers clustered closely on the phylogenetically tree but in a separate clade from the community of viruses.

Overall, this is a very good paper.  The authors have done excellent work and the Discussion and Conclusions verify the data.  It is important that this type of study be published as it confirms the zoological nature of SARS CoV2 and the fact that there is likely mink to mink, mink to human and human to mink transmission.  It is interesting that there have few if no cases of SARS CoV2 in mink farms in 2022-2023.

Author Response

RESPONSE: We thank the reviewer for recognizing the importance and the significance of our manuscript.

Reviewer 2 Report

Comments and Suggestions for Authors

Dear Authors,

Thank you very much for your valuable contribution to understanding the zoonotic transmission of  SARS-CoV-2. I’ve found your manuscript interesting and informative.

Please pay attention to using the proper names of molecular techniques, like PCR, Real-Time PCR (qPCR), RT-PCR, and RT-qPCR. These abbreviations are commonly confused, even in scientific literature. The “RT-PCR” should only be used to describe reverse transcription PCR and not the real-time PCR [Bustin, 2009]. Herein, since RNA was your matrix, reverse transcription was used to generate cDNA, and further quantitative amplification was applied; the proper name should be quantitative reverse transcription polymerase chain reaction (RT-qPCR or qRT-PCR) or real-time reverse transcription PCR (abbrev also RT-qPCR or qRT-PCR).

Bustin, S. A., Benes, V., Garson, J. A., Hellemans, J., Huggett, J., Kubista, M., Mueller, R., Nolan, T., Pfaffl, M. W., Shipley, G. L., Vandesompele, J., & Wittwer, C. T. (2009). The MIQE guidelines: minimum information for publication of quantitative real-time PCR experiments. Clin Chem, 55(4), 611-622. https://doi.org/10.1373/clinchem.2008.112797

Author Response

RESPONSE: We thank the reviewer for recognizing the importance and the significance of our manuscript. We agree with the reviewer and have changed to RT-qPCR throughout the manuscript.

Reviewer 3 Report

Comments and Suggestions for Authors

This paper describes well an important aspect of zoonotic disease transmission. Intensive farming of a species that can be infected by potential human pathogens requires stringent biosecurity controls and this paper emphasises this with the intervention described, and highlights the consequences (commercial but also animal and human health) of not having those controls in place. I just have a few minor questions.

Lines 127-128. The paper does not explain why half of the mink sampled were done so sedated and half unsedated. The results suggest no effect on the sample, so is this for animal welfare reasons? If so, which is the better route to go? From human health and safety I would think handling an unsedated mink would not be my choice!   

Lines 149-150. Why were swabs taken from wildlife stored in different medium - BHI as opposed to VTM for those from mink and staff? Brief explanation would be useful.

Lines 278 - 292. In terms of the staff questionnaire, did you ask anything about hand hygiene eg, access to handwash facilities, training etc. Although airborne transmission is the major route hand to oro-nasal cannot be ruled out. On a similar topic, lines 301 - 302 refers to use of gloves did that include training on safe glove removal without cross contamination, and disposal?

Line 366 says "either staff or visitors" - this might be a throwaway line but opens up another topic.  There is no information in the paper about visitors - who might they be? would they come into contact with animals and if so could they have been the source of infection? What biosecurity arrangements were there for visitors then and what are they now? This is well established practice for other farming environments.

Lines 390 - 391 says "cannot rule out" ...."were not infected". Please clarify - did you mean "could have been infected by the same..."

Author Response

This paper describes well an important aspect of zoonotic disease transmission. Intensive farming of a species that can be infected by potential human pathogens requires stringent biosecurity controls and this paper emphasises this with the intervention described, and highlights the consequences (commercial but also animal and human health) of not having those controls in place. I just have a few minor questions.

RESPONSE: We thank the reviewer for recognizing the importance and the significance of our manuscript.

Lines 127-128. The paper does not explain why half of the mink sampled were done so sedated and half unsedated. The results suggest no effect on the sample, so is this for animal welfare reasons? If so, which is the better route to go? From human health and safety I would think handling an unsedated mink would not be my choice!   

RESPONSE: Blood samples could be collected only from sedated mink. We limited the number of mink that were sedated because, a) the producer expressed concerns over complications associated with sedation (all mink recovered well from sedation, however), and b) sedating mink is very time consuming, and limiting the numbers allowed investigators to sample more mink for active respiratory infection. We’ve now included this some rationale for these decisions in lines 120 and 128. Of note, we do not draw too much attention to the that results of positivity being unaffected by awake or sedated sampling, because investigators noted the mink on these farm were unusually docile and therefore these findings may not be applicable to other farmed mink.

Lines 149-150. Why were swabs taken from wildlife stored in different medium - BHI as opposed to VTM for those from mink and staff? Brief explanation would be useful.

RESPONSE: This is an astute observation. Wildlife samples were stored in BHI only because the field team collecting wildlife samples was separate from the field teams collecting mink, companion animal, and human samples. This was their preferred medium. Since these two media are generally considered equivalent for SARS-CoV-2 and we do not think this affected methods or results, we do not feel it necessary to add additional detail.

Lines 278 - 292. In terms of the staff questionnaire, did you ask anything about hand hygiene eg, access to handwash facilities, training etc. Although airborne transmission is the major route hand to oro-nasal cannot be ruled out. On a similar topic, lines 301 - 302 refers to use of gloves did that include training on safe glove removal without cross contamination, and disposal?

RESPONSE: Thank you for noticing the training we provided was not clear. We have now included a line stating, “training included cleaning and disinfection best practices, how to don and doff PPE, and hand hygiene.”

Line 366 says "either staff or visitors" - this might be a throwaway line but opens up another topic.  There is no information in the paper about visitors - who might they be? would they come into contact with animals and if so could they have been the source of infection? What biosecurity arrangements were there for visitors then and what are they now? This is well established practice for other farming environments.

RESPONSE: This is a good point. During our interviews, employees did not disclose visitors in the month before or after the outbreak, so we have removed mention of visitors from this line. Because of the targeting mink farms experience from animal rights organizations, mink farms, unlike other farming operations, are closed to the public and prefer to operate in anonymity. The only visitors we speculate they receive might be friends or family members of the staff, although, as we mention, they did not report visitors during the outbreak period.  

Lines 390 - 391 says "cannot rule out" ...."were not infected". Please clarify - did you mean "could have been infected by the same..."

RESPONSE: Thank you. This sentence now reads, “However, our epidemiologic evidence cannot rule out the possibility that these COVID-19 positive staff were infected by the same asymptomatic individual that exposed the mink” to avoid a double negative.

Reviewer 4 Report

Comments and Suggestions for Authors

Zoonotic transmission is clearly of significant public interest. The  comprehensive introduction in this paper describes cases of COVID infections in mink where large-scale culling was carried out in Denmark to prevent spread. The authors provide an interesting and comprehensive account of animal to human transmission of SARS-CoV-2 citing Tan et al (2022) who described transmission of the virus from humans to animals and potential adaptation in an animal host.

Very comprehensive outline of methodology employed. Sample collections were carried out with a clear plan enabling accurate tracking and time-line for COVID infections in mink and staff of the facility.

The methodology is documented in detail with useful sub-headings enabling the reader to navigate around the paper.

The wildlife and human sample testing protocols follow published methods and the analytical methods, including genomic analyses are well chosen and clearly carefully conducted. 

Results are clearly presented in Tables 1 and 2 with additional comments in text relating to the testing of companion animals owned by farm employees. Descriptions of the routine actions of farm employees provide the reader with a mental picture of day-to-day operations of the mink farm. It was interesting to read  (Table 3) that two spike mutations detected in other mink farm outbreaks in the Netherlands were present in all mink sequences in this study. They do mention, later in discussion, that an asymptomatic visitor may have introduced the virus to this mink farm.

Fig 2 and Fig 3 together provide a nice pictorial view of the results from the Nextstrain divergence tree and a detailed view of the mink clade relative to the Michigan human community sample sequences.

Of note was the observation that the Nextstrain molecular clock estimation indicated a faster pace of evolution within the clade than in the broader Michigan community!

As for the other sections of this paper the discussion also provides a dynamic discussion of the results and the possible impact of the new findings. The results do provide good support for the idea that SARS-CoV-2 ad in particular lineages circulating in 2020, are highly contagious to mink and that where there is intensive farming of mink at high densities and in close proximity to each other, highly contagious environments are created resulting in widespread and rapid transmission. Transmission to other susceptible species including humans is then highly likely.

The author contributions to this impressive study reveal a team approach reflected in the diagnostic, analytical and interpretive skills resulting in this fine paper that significantly advances our knowledge of zoonotic spread of SARS-CoV-2.

Minor suggestion

L 226 "In order.." can be deleted it really adds nothing to the sentence.

Author Response

RESPONSE: We thank the reviewer for recognizing the importance and the significance of our manuscript. We have deleted “In order” per the reviewer suggestion.